# High-Frequency Audiometry for Early Detection of Hearing Loss: A Narrative Review

**DOI:** 10.3390/ijerph18094702

**Published:** 2021-04-28

**Authors:** Michaela Škerková, Martina Kovalová, Eva Mrázková

**Affiliations:** 1Department of Epidemiology and Public Health, Faculty of Medicine, University of Ostrava, 703 00 Ostrava, Czech Republic; martina.kovalova@osu.cz (M.K.); orl.centrum@centrum.cz (E.M.); 2Center for Hearing and Balance Disorders, 708 00 Ostrava, Czech Republic; 3Department of ENT, Regional Hospital Havířov, 736 01 Havířov, Czech Republic

**Keywords:** audiometry, high-frequency audiometry (HFA), hearing loss, hearing test, hearing threshold, noise-induced hearing loss (NIHL), age-related hearing loss, occupational noise, recreational noise, noise exposure

## Abstract

The WHO considers hearing loss to be a major global problem. A literature search was conducted to see whether high-frequency audiometry (HFA) could be used for the early detection of hearing loss. A further aim was to see whether any differences exist in the hearing threshold using conventional audiometry (CA) and HFA in workers of different age groups exposed to workplace noise. Our search of electronic databases yielded a total of 5938 scientific papers. The inclusion criteria were the keywords “high frequency” and “audiometry” appearing anywhere in the article and the participation of unexposed people or a group exposed to workplace noise. Fifteen studies met these conditions; the sample size varied (51–645 people), and the age range of the people studied was 5–90 years. Commercial high-frequency audiometers and high-frequency headphones were used. In populations unexposed to workplace noise, significantly higher thresholds of 14–16 kHz were found. In populations with exposure to workplace noise, significantly higher statistical thresholds were found for the exposed group (EG) compared with the control group (CG) at frequencies of 9–18 kHz, especially at 16 kHz. The studies also showed higher hearing thresholds of 10–16 kHz in respondents aged under 31 years following the use of personal listening devices (PLDs) for longer than 5 years. The effect of noise-induced hearing loss (NIHL) first became apparent for HFA rather than CA. However, normative data have not yet been collected. Therefore, it is necessary to establish a uniform evaluation protocol accounting for age, sex, comorbidities and exposures, as well as for younger respondents using PLDs.

## 1. Introduction

The monitoring of hearing functions in the context of the International Classification of Functioning, Disability, and Health has revealed that hearing loss is a major global problem [1]. The World Health Organization (WHO) speaks of hearing loss as an epidemic of the 21st century [2]. More than 1 billion young people (aged 12–35 years) are at risk of hearing loss due to recreational exposure to intense sound [3]. Hearing loss is an invisible and very stigmatizing chronic disease. In terms of severity, hearing loss is second only to mental disability. It is the most common and the most serious human sensory defect. The incidence of hearing loss is continuously increasing and, at the same time, as with other diseases, prevention and early detection of hearing loss can lead to earlier provision of appropriate care. This could significantly lower the incidence of damage and result in better quality of life [2,3]. In its 2017 report, the Lancet Commission on dementia prevention, intervention and care found that 35% of dementia cases are attributable to a combination of nine modifiable risk factors, divided into early-life, mid-life and late-life risk factors. The most significant among the mid-life (45–65 years of age) risk factors is hearing loss, as 55 is the mean age at which the presence of hearing loss is shown to increase the risk of dementia. Despite being classified as a mid-life risk factor, evidence suggests that it continues to increase the dementia risk in later life. Although the report makes it clear that the exact mechanism through which hearing loss affects dementia risk is not fully understood, the authors cite studies suggesting that hearing loss might either add to the cognitive load of a vulnerable brain, leading to changes in the brain, or lead to social disengagement or depression and accelerated brain atrophy, all of which could contribute to an accelerated cognitive decline. Moreover, given the importance of language in humans (a key contributor to the coevolution of social interaction and a larger brain size), hearing loss could lead to uniquely interrelated and detrimental social, cognitive and brain effects [4]. When not detected, affected individuals gradually disengage from society and suffer from the accompanying consequences, such as loss of employment, loss of interpersonal communication and the associated intensification of social isolation and dementia. According to the WHO, the global prevalence of hearing loss in 2018 was 466 million people. Of these, 432 million were adults with impaired hearing. This number continues to increase as the average age of the world’s population increases. According to WHO projections, the number of people suffering from hearing loss is expected to reach 630 million by 2030 and 900 million by 2050 [2,3]. The auditory system gradually deteriorates with age, starting as early as the age of 40. The WHO states that approximately one-third of the population above the age of 65 suffers from hearing loss. The WHO cites the hear-it report from the Hearing Loss Association of America showing a hearing loss prevalence of 30% in people aged over 65 years of age [5,6]. The same report found that almost all people over 80 years of age have some degree of hearing impairment. Worsened hearing in the elderly is mainly associated with a reduced ability to comprehend speech, especially in noisy environments or in places with background noise. People with impaired hearing realize that someone is talking to them, but they are unable to understand what is being said [7,8,9].

Conventional pure tone audiometry (CA) is currently one of the most widely used methods for the diagnosis of hearing impairment. CA operates over the 0.125–8 kHz frequency range. High-frequency audiometry (HFA) is used to examine the hearing threshold in the frequency range of 8–20 kHz. These are the frequencies at which damage to the hearing threshold can first be observed [10]. HFA has been studied for several decades, but the lack of commercially available equipment and the standardization of calibration recommendations have long limited its use. Specially adapted audiometers capable of generating tones with frequencies of up to 20 kHz are used for the test. Experimental studies have established rules and conditions for testing [11,12]. Over the last 20 years, the development of new methods and devices and the increased penetration of computers in both daily life and healthcare have enabled HFA to increasingly become the standard. The current applicable standard, “IEC 60645-1 Electroacoustics—Audiometric equipment, Part 1: Equipment for pure-tone and speech audiometry”, specifies the general requirements for audiometers designed for use in determining hearing threshold levels relative to the standard reference threshold levels, established by means of psychoacoustic test methods and those designed to perform psychoacoustic tests using speech material. The objective of this standard is to ensure that (1) hearing tests in the frequency range of 0.125–16 kHz conducted on a given human ear and performed with different pure-tone audiometers that comply with this standard give the same results; (2) the results obtained represent a valid comparison between the hearing of the ear tested and the reference threshold of hearing; and (3) the means for presenting speech material to a subject in a standardized manner are provided. This ensures that hearing tests using a specific speech signal and a specific manner of signal presentation, when performed with different audiometers that comply with this standard, give very similar results. Furthermore, the standard classifies audiometers according to the range of test signals they present, the mode of operation or their presumed primary application [13]. Conventional pure tone audiometry along with HFA and equipment (headphones) are used according to the standards of EN ISO 8253-1:2010 Acoustics—Audiometric test methods and EN ISO 266:1997 Acoustics—Preferred frequencies [14,15]. These standards were last reviewed and confirmed in 2018 and 2021. Therefore, these versions remain current. HFA is currently not used in common practice to determine the presence of hearing impairment [12]. Some experimental studies have used electrophysiological examinations to monitor the effect of hearing damage due to noise exposure. Close monitoring of the outer hair cell function when hearing thresholds are clinically normal could provide a timely measure of noise-induced hearing damage, especially for individuals with high levels of noise exposure [16].

In this study, a literature search was conducted to find out whether HFA can be used for early hearing loss detection. Furthermore, we set out to find out whether there are differences in the hearing threshold, as shown using CA and HFA, among different age groups between workers who are exposed to noise in their workplaces and unexposed people.

## 2. Materials and Methods

### 2.1. Search Strategy

The literature search was conducted from 15 November 2020 to 20 January 2021 in the PubMed, Scopus and Web of Science electronic databases. The search strategy combined key words related to HFA.

### 2.2. Inclusion Criteria

The criteria for the selection of publications were as follows:The keywords “high frequency” and “audiometry” should appear somewhere in the text of the article;The date of publication is between 2000 and 2020, in line with the time frame determined by the standards IEC 60645-1; 2017, EN ISO 8253-1:2010 and EN ISO 266:1997 [13,14,15];The study includes either an unexposed population or a group exposed to workplace noise;The results are evaluated with statistics;It is possible to compare the hearing thresholds for CA and at least 5 high frequencies in the range of 9–20 kHz;The study is written in either English, Portuguese or Spanish.

### 2.3. Exclusion Criteria

All duplicate articles were excluded, as well as studies with unclear exposure conditions. Conference papers, letters and book chapter categories were excluded. Articles concerning hearing loss in association with ototoxic drugs, tinnitus or diabetes were also excluded.

## 3. Results

The search yielded a total of 5938 scientific publications. A total of 5855 publications that did not directly relate to the examined issues were excluded. Finally, 83 articles were selected, of which 30 were duplicated in three databases. The remaining 53 articles concerned HFA, but not all of them were relevant to our search, as 26 of them concerned hearing loss in association with ototoxic drugs, tinnitus or diabetes. Of the remaining 27 articles, 15 papers that met all six conditions for inclusion in our study were selected. Figure 1 presents a step-by-step representation of our screening process.

Table 1 and Table 2 summarize the characteristics of the 15 studies included. Studies were conducted in Brazil (*n* = 4), Iran (*n* = 2), Italy (*n* = 2), Spain (*n* = 1), Saudi Arabia (*n* = 1), India (*n* = 1), China (*n* = 1), Greece (*n* = 1) and the USA (*n* = 1), and there was also a systematical review comprising multiple countries around the world (*n* = 1).

In this review, the studies were evaluated by the following parameters: the number of involved participants; the differences in hearing thresholds between an unexposed population and a population exposed to noise; the correlation between hearing thresholds, assessed by HFA and age, especially focused on particular age groups; the difference in hearing thresholds if the age groups defined by the studies had non-identical age ranges; and the similarity in hearing thresholds if the groups had identical age ranges.

The number of participants in each study ranged from 51 to 645. For the years 2000–2020, 5 articles were published in 2000–2010, 10 in 2011–2020, 5 in 2015–2020 and 2 in 2017–2020. The articles were divided into two groups. The first group comprised studies on people who were not exposed to workplace noise (Table 1). Two of these studies were focused on users of PLDs, one of the newly examined risk factors for hearing loss. The second group comprised studies on people who were exposed to noise in their workplace (EG), along with a control group (CG) without noise exposure (Table 2).

As shown in Table 1, the sample size varied from 51 to 645 participants in studies that were done on unexposed individuals. The age of the subjects varied from 5 to 90 years, depending on the study. Grouping based on age also varied, with all studies including both men and women. Rodríguez Valiente et al. studied 645 people who were evenly distributed among 10-year age groups ranging from 5 to 90 years, with 90 respondents in each group [17]. Other studies followed respondents in the age group of 18–31 years, with age intervals ranging from 11 to 15 years [18,19,20]. Oppitz et al. [21] used the largest age range. Kumar et al. and Le Prell et al. monitored the effect of personal listening devices (PLDs) [19,20]. Each study used a different type of audiometer [12,13,14,15,16]. Sennheiser headphones were used in four cases, and Rodríguez Valiente et al. and Le Prell used the Sennheiser HDA200 [17,18,19,20].

As shown in Table 2, studies in which people were exposed to noise in the workplace involved 40–282 subjects. The control groups had 32–142 participants. Five of the studies involving workplace noise exposure only assessed men [22,23,24,25,26], four of them included both men and women [27,28,29,30], and one study was a review [31]. The age range was 15–61 years, and the age groups used also varied. All studies compared CA and HFA, and five of them even included a frequency of 20 kHz. An Interacoustic audiometer was used in four studies; three of these used the AC 40 type [22,26,28], and one used the AS10HF type [31]. Two studies used a Madsen audiometer [23,30], a further two used the Amplaid [22,28], and one study used a Labat Audiopack [27]. Sennheiser headphones were used in five studies; four of them used the HDA 200 [23,24,25,30], and one used the HDA 500 [27]. Three studies used Koss headphones [22,26,28]. The systematic review conducted by Antonioli et al. compared six studies with six different audiometers and different types of headphones [31].

**Table 1 ijerph-18-04702-t001:** Overview of the selected articles on hearing loss and high-frequency audiometry in populations not exposed to workplace noise.

Author, Year	Number of Respondents and Study Design	Age Range (Years) and Groups	City, Country	Audiometry and Frequency Range (kHz)	Audiometer Type	Headphones	Objective	Findings
Rodríguez Valiente et al., 2014 [17]	645 (321 men and 324 women). No workplace noise exposure. Prospective study.	5–90 Age groups: 5–19, 20–29, 30–39, 40–49, 50–59, 60–69, 70–90. Same number in each group. Ears not distinguished. Divided by gender.	Madrid, Spain	CA 0.125–8 HFA 8–20	Madsen Orbiter 922, version 2, Madsen Electronics	CA: Telephonic TDH-39P HFA: Sennheiser 200	Determine threshold values over the 0.125–20 kHz range in healthy, professional, unexposed people; try to set new standards.	In the group of people aged 20–69 years, the hearing threshold values were lower in women than in men, especially at 12.5 and 16 kHz.
Oppitz et al., 2017 [21]	60 (11 men and 49 women). No workplace noise exposure. Cross-sectional, prospective study.	18–58 Age groups: 18–30, 31–58. Group 1: 49 people. Group 2: 11 people. Left and right ears. Not divided by gender.	Santa Marie, Brazil	CA 0.250–8 HFA 9–18	Interacoustics AS10HF	CA: Telephonics TDH-39P HFA: KOSS R/80	Evaluate high-frequency hearing thresholds and try to compare differences between the ears; verify correlation between hearing quality and aging.	There was a progressive increase in hearing thresholds above 14 kHz. The increased hearing thresholds were found in both ears and were proportional to the rising frequency and age.
Barbosa de Sá et al., 2007 [18]	51 (19 men and 32 women). No workplace noise exposure. Cross-sectional, prospective study.	18–29 1 age group. Left and right ears. Divided by gender.	Rio de Janeiro, Brazil	CA 0.250–8 HFA 8–18	Amplaid 460	CA: Telephonics 236D 100-1 HFA: Sennheiser HD 520 II	Analyze results related to high-frequency hearing thresholds in individuals aged 18–29 years without otological problems.	There were no significant differences in hearing thresholds between men and women aged 18–29 years. Significant differences in hearing thresholds between the left and right ears were found only at 11–12 kHz. Over 16 kHz, hearing thresholds increased bilaterally.
Kumar et al., 2016 [19]	100 30 (10 men and 20 women), 70 people using PLDs (22 men and 48 women). Study design unknown.	15–30 1 age group. Ears not distinguished. Not divided by gender.	New Delhi, India	CA 0.125–8 HFA 9–20	Labat Audiolab Audiometer	CA/HFA: Sennheiser HDA 300	Examine changes in HFA hearing thresholds in PLD users and compare them with an unexposed group.	Using a PLD for more than 5 years at a high volume led to significantly increased hearing thresholds at 3, 10 and 13 kHz.
Le Prell et al., 2013 [20]	87 (34 men and 53 women) using PLDs. Retrospective analysis.	18–31 1 age group. Left and right ears. Divided by gender.	Florida, USA	CA 0.250–8 HFA 10–16	Grason-Stadler model 61 (GSI 61)	CA: EAR 3A HFA: Sennheiser HDA200	Determine whether HFA thresholds for university students differ depending on exposure to recreational noise.	Subjects who used a PLD over the long term (5 years or more) showed statistically significant threshold differences (3–6 dB higher) at the highest frequencies tested (10–16 kHz).

HFA = high-frequency audiometry; CA = conventional pure tone audiometry; and PLDs = personal listening devices.

**Table 2 ijerph-18-04702-t002:** Overview of the selected articles on hearing loss and high-frequency audiometry in populations exposed to workplace noise.

Author, Year	Number of Respondents and Study Design	Age Range (Years) and Groups	City, Country	Audiometry and Frequency Range (kHz)	Audiometer Type	Headphones	Objective	Findings
Maccá et al., 2014 [27]	24 EG ultrasound (2 men and 22 women), 113 EG (93 men and 20 women) 148 CG (62 men and 86 women). Study design unknown.	15–59 Age groups: 15–19, 20–29, 30–39, 40–49, 50–59. Ears not distinguished. Not divided by gender.	Padua, Italy	CA 0.125–8 HFA 9–18	Labat Audiopack audiometer	CA: Standard headphones HFA: Sennheiser, HD 500	Investigate the effects of age, ultrasound and noise on high-frequency hearing thresholds.	After stratification for age, there was a significantly higher hearing threshold in EG than CG at 9–10 and 14–15 kHz only for those under 30 years of age.
Mehrparvar et al., 2014 [22]	142 EG 121 CG Only men. Cross-sectional, prospective study.	<50 1 age group. Left and right ears. Only men.	City unknown, Iran	CA 0.5–8 HFA 10–16	Interacoustic AC40	CA: TDH 39 HFA. Koss R/80	Compare three methods of assessing hearing loss due to noise (HFA, CA, DPOAE)	The most commonly affected frequencies with statistically significantly higher hearing thresholds in EG compared with CG were 4 and 6 kHz in CA and 14 and 16 kHz in HFA. HFA was the most sensitive test for detection of hearing loss in workers exposed to >85 dBA noise.
Mehrparvar et al., 2011 [28]	120 EG (108 men and 12 women) 120 CG (106 men and 14 women). Historic cohort.	<50 1 age group. Left and right ears. Not divided by gender.	City unknown, Iran	CA 0.250–8 HFA 10–16	Interacoustic AC40	CA: TDH 39 HFA: Koss R/80	Compare thresholds with both CA and HFA in both ears in exposed and unexposed individuals to assess the efficiency of the methods when revealing hearing loss.	Statistically significantly higher mean hearing thresholds in EG compared with CG were found at 4, 6 and 16 kHz, with the most significant differences found at 16 kHz in both ears.
Ma et al., 2018 [23]	134 EG 101 CG Only men. Cross-sectional study.	20–59 Age groups: 20–29, 30–39, 40–49 50–59. Ears not distinguished. Only men.	City unknown, China	CA 0.250–8 HFA 9–20	Madsen Conera	CA: TDH–39 HFA: Sennheiser HDA 200	Investigate the usefulness of HFA as an assessment test of the hearing statuses of civilian pilots.	Statistically significantly higher mean hearing thresholds in EG compared with CG were found at most of the high frequencies tested. In particular, the largest differences between hearing thresholds were found at 16 kHz for subjects aged 20–29 and 30–39, at 12.5 kHz for those aged 40–49 years old and at 10 kHz for those aged 50–59 years old.
Ahmed et al., 2001 [24]	187 EG 52 CG Only men. Cross-sectional study.	Undefined–44 Age groups: <25, 25–34, 35–44. Ears not distinguished. Only men.	City unknown, Saudi Arabia	CA 0.250-8 HFA 10–18	Interacoustics AS10HF	CA: Koss HV-1A HFA: TDH-50P	Investigate the reliability and effects of age and noise on HFA hearing thresholds.	A multivariate analysis showed that the primary indicator of the hearing threshold at high frequencies is age, and noise exposure is a secondary predictor of hearing thresholds at high frequencies (10–18 kHz).
Somma et al., 2008 [25]	84 EG 98 CG Only men. Study design unknown.	21–60 Age groups: 21–30, 31–40, 41–50, 51–60. Ears not distinguished. Only men.	City unknown, Italy	CA 0.250-8 HFA 9–18	Amplaid A319, Amplifon	CA: TDH-49 HFA: Sennheiser HDA 200	Compare HFA and CA to assess thresholds among workers exposed to workplace noise.	Statistically significantly higher hearing thresholds between EG and CG were found for those aged 21–30 years old at all frequencies (9–18 kHz) and for those aged 31–40 years old at frequencies of 9–14 kHz.
Korres et al., 2008 [29]	139 EG (68 men and 53 women) 32 CG (18 men and 14 women). Study design unknown.	24–55 1 age group. Left and right ears. Not divided by gender.	City unknown, Greece	CA 0.250-8 HFA 9–20	Amplaid 321, Twinchannel	CA: TDH-49 HFA: Sennheiser HDA 200	Evaluate hearing in industrial workers exposed to workplace noise using CA and HFA and compare it with CG.	Statistically significantly higher hearing thresholds between EG and CG were found at 4–18 kHz, especially at 12.5–18 kHz. A statistically significant correlation between an increased duration of exposure and higher hearing thresholds was found at all frequencies except for 10 kHz.
Rocha et al., 2010 [26]	47 EG 33 CG Only men. Cross-sectional, retrospective cohort study.	30–49 Age groups: 30–39, 40–49. Ears not distinguished. Only men.	Rio de Janeiro, Brazil	CA 0.250-8 HFA 9–20	Interacoustic AC40	CA: TDH-39P HFA: Koss HV/PRO	Analysis of HFA results in people exposed to noise with normal results for CA.	The EG had a statistically significantly higher hearing threshold than CG at 16 kHz in participants aged 30–39 years. The results were most significant in the 40–49 years age group, where EG showed significantly higher hearing thresholds than CG at 14 and 16 kHz.
Goncalves et al., 2015 [30]	40 EG (10 men and 32 women) CG 40 Historic cohort study.	23–61 1 age group. Left and right ears. Divided by gender.	Curitiba, Brazil	CA 0.5–8 HFA 9–16	Madsen Itera II, GN Otometrics	CA: Standard HFA: Sennheiser HDA 200	Use HFA to evaluate hearing among dentists exposed to workplace noise for varying durations.	Statistically significantly higher hearing thresholds in EG compared with CG were observed at 0.5, 1, 6 and 8 kHz in the right ear. No differences were observed between the EG and CG for high frequencies.
Antonioli et al., 2016 [31]	Exposed workers and unexposed people. Both genders. Systematic review, meta–analysis.	18–60 Different age groups. Ears not distinguished. Not divided by gender. 6 studies	Many countries	CA 0.250–8 HFA 10–20	Interacoustics AS10HF; Amplaid A3 19; Amplaid A321; Interacoustic AC 40; Siemens SD50; GSI 61	HFA: Koss R/80; HDA200; R80; HDA200; HD 200; TDH-39p	Retrospective and secondary systematic revision of publications using HFA to monitor the hearing of workers exposed to workplace noise.	At 16 kHz, HFA is sensitive enough for the early detection of hearing loss. This is true for 4 kHz as well, but the outcome is not as significant. Further studies are therefore needed to confirm the importance of HFA for the early detection of hearing loss in people exposed to workplace noise.

HFA = high-frequency audiometry; CA = convectional pure tone audiometry; NIHL = noise-induced hearing loss; EG = group exposed to workplace noise; CG = control group without exposure to workplace noise; and DPOAE = distortion product otoacoustic emissions.

### 3.1. Studies with People Not Exposed to Workplace Noise

Articles monitoring high-frequency hearing loss in unexposed individuals reported an increase in hearing thresholds with an increasing frequency as well as with increasing age [17,18,21]. In populations unexposed to workplace noise, significantly higher hearing thresholds were found at frequencies of 14 and 16 kHz, and these increased with age [17,20]. In a representative cohort of 645 people, it was found that those under 40 years were able detect sounds at frequencies of up to 18 kHz, people between 40 and 49 could detect sounds at frequencies of up to 14 kHz, and those older than 50 could only detect sounds at frequencies of up to 11.2 kHz. The mean hearing thresholds at each frequency (11.2 to 20 kHz) were lower in women than men. Statistically significantly higher hearing thresholds were found in men than in women at 12.5 and 16 kHz in the 20–29 year group, at 16 kHz in the 30–39 year group, at 11.2 and 16 kHz in the 40–49 year group, at 10 kHz in the 50–59 year group and at 12.5 and 18 kHz in the 60–69 year group. Nevertheless, a comparison of different studies showed that the hearing thresholds for men and women were similar [17]. This was confirmed by Barbosa de Sá et al., who found no significant difference in hearing thresholds between men and women aged 18–29 years [18]. Oppitz et al. found higher hearing threshold values in the right ear than in the left ear at 10, 11 and 14 kHz. At frequencies above 14 kHz, the progressive increase was proportional to the frequency and was bilateral (binaural), with threshold values increasing with age at all frequencies [21]. Barbosa de Sá et al. found that the thresholds were similar in the left and right ears, with significant differences between the ears only being observed at 11 and 12 kHz, with the right ear being worse [18].

Kumar et al. found that high frequency thresholds could be used for the early detection of noise-induced hearing loss (NIHL) in PLD users. It was found that those using PLDs at high volumes for less than 5 years showed no significant differences in hearing threshold values. In contrast, using a PLD for more than 5 years led to significantly increased hearing thresholds at 3, 10 and 13 kHz [14]. Le Prell et al. found statistically significant threshold differences (3–6 dB higher) at 10–16 kHz in respondents who had used a PLD for a long period of time or at a very high volume after just 5 years [20]. HFA could be used for the early detection of NIHL in PLD users [19,20]. Both studies examined the age category of below 31 years. This generation generally has more frequent usage of mobile phones, headphones and other PLDs.

Specific threshold analyses focused on risk stratification of permanent hearing loss according to clearly defined levels of exposure to music through PLDs are missing. This is due to many restrictions of conducting such research that are related to the long latency time from exposure to effect, the problems to correctly estimate the exposure and the lack of sensitive measures to detect early signs of hearing loss [32].

### 3.2. Studies on Workplace Noise Exposure

Age was found to be the primary predictor and noise exposure the secondary predictor of an increase in high-frequency hearing thresholds. The results suggest that HFA may be useful for the early diagnosis of noise-induced hearing loss, especially for younger groups of workers (up to 30 years of age) [22,23,24,25,26,27,28,29,30].

HFA is more sensitive for NIHL detection than CA; it can be used for early diagnosis of hearing loss, and thus it can contribute to the prevention of hearing loss, even at lower frequencies, especially at frequencies used for speech [23,25,28]. Studies comparing the EG and the CG found that in those aged over 30 years, hearing loss was apparent in HFA before it became apparent in CA [23,25,27]. Studies suggest that HFA rather than CA could be useful as an early probe for hearing loss resulting from noise [24]. In a sample of working age respondents, Korres et al. showed that HFA is a useful addition to CA for the audiological evaluation of people exposed to workplace noise [29]. Mehrparvar et al. compared the following methods: CA, distortion product otoacoustic emissions and HFA. HFA was confirmed as the most sensitive method for detecting hearing loss from hazardous noise exposure in the workplace [22]. Maccá et al. stated that hearing loss at high frequencies is affected not only by age, but also by the duration of exposure to a noisy work environment [27]. This was confirmed by Goncalves et al., who showed that dentists working in surgeries for more than 10 years had significantly greater hearing loss at high frequencies compared with those in the control group. Eighty-one percent of dentists were not informed of the risk of noise during their university studies, and 15% of dentists (EG) had sensorineural hearing impairment, while in the CG, the frequency of occurrence was 2.5% [30]. Korres et al. found a statistically significant correlation between the duration of exposure and the hearing threshold at all frequencies [29]. Changes in high-frequency thresholds were found to be accelerated by noise exposure in the first few years, suggesting that HFA could be a useful preventative measure for monitoring younger workers exposed to workplace noise [25]. Statistically significant differences were found between people exposed to noise for <10 years and a control group at 2–8 kHz, 9–10 kHz and 14–15 kHz in people aged up to 39 years of age. Age is a secondary factor to hearing loss; at conventional frequencies, the hearing threshold increases after the age of 20. After stratification for age, significant differences between the EG and the CG were found at 9–10 kHz and 14–15 kHz only for those who were aged below 30 years [27]. Larger differences were found between the exposed and control groups at 4–18 kHz, and these were more evident at 12.5–18 kHz. There was a statistically significant correlation between the differences in thresholds and duration of exposure at all frequencies (0.25–20 kHz), except for 10 kHz [29]. HFA performs well in the frequency range of 12.5–18 kHz, but there is greater variability in the results compared with CA [29]. The results of Rocha et al. were the most significant for the 40–49-year age group, and the exposed group showed significantly higher thresholds than the control group at 14 and 16 kHz. The EG of individuals aged 30–39 years showed a significantly higher threshold than CG only at 16 kHz [26]. According to Ma et al., the most frequently affected frequencies for subjects aged 30–39 years are 14 and 16 kHz [23]. A stepwise regression analysis showed that in 21–40-year-old workers, the effect of noise was apparent with both CA and HFA, while in older people, the noise effect was only apparent at frequencies of up to 6 kHz, and the impact of age was significantly higher at higher frequencies [25]. Mehrparvar et al. stated that the most frequently affected frequencies were 4 and 6 kHz for CA and 14 and 16 kHz for HFA [22]. They also found abnormal hearing in 29% of participants exposed to workplace noise with CA, 69% with HFA, 22% with low-frequency DPOAE and 52% with high-frequency DPOAE [22].

In another study comparing the EG and the CG, Mehrparvar et al. found that in the EG, the highest hearing thresholds were observed at 4 kHz in the left ear, 6 kHz in the right ear and 16 kHz when both ears were tested. The hearing threshold was significantly higher at 16 kHz than at 4 kHz. There was no statistically significant difference between the right and left ears in either group. Hearing loss was more common in males than females, but the difference was not statistically significant [28]. Ma et al. found differences in thresholds between the EG and the CG in all age categories [23]. Compared with the CG, the mean threshold at frequencies of 0.25–20 kHz in the EG increased by 3.15 dB for CA and by 7.8 dB, respectively, for HFA for the 20–29-year age group; by 2.4 and 9.9 dB, respectively, for the 30–39-year age group; by 3.8 and 8.2 dB, respectively, for the 40–49-year age group; and by 10.8 and 16.9 dB, respectively, for the 50–59-year age group [23]. Additionally, Ma et al. found that the results for particular frequencies were more sensitive to noise than other frequencies: 14 and 16 kHz for the 20–29-year age group, 11.2 and 12.5 kHz for the 40–49-year age group and 11.2 and 10 kHz for the 50–59-year age group. Significant differences in HFA were also observed between the EG and the CG with normal CA hearing thresholds. Seventy-five percent of pilots were shown to have hearing loss in at least one ear and at least one frequency with CA; the corresponding proportion was 95% with HFA [23].

Statistically significantly higher hearing thresholds in the EG than the CG were found at all frequencies (9–18 kHz) for the 21–30-year age group and at frequencies of 9–14 kHz for those aged 31–40 years old. On the other hand, in the 41–50-year age group, statistically significantly higher hearing thresholds were found in the EG compared with the CG only at 12.5 and 14 kHz, while no significant differences were found for the oldest workers aged 51–60 years old [25].

Antonioli et al. selected over 6000 articles for a retrospective and systematic review of the use of HFA for the monitoring of hearing loss of those exposed to workplace noise [31]. Of these, only six articles met the criteria for the study, and the difference between exposed and unexposed persons was seen mainly at 16 kHz, with a slightly smaller difference at 4 kHz. Antonioli et al. conducted a meta-analysis that clearly determined the hearing threshold for each frequency (i.e., 2, 3, 4, 6, 8, 10, 14 and 16 kHz). They evaluated the mean, standard deviation, median and 95% CI for the EG and the CG, but they did not take age groups into account, and these differed among the studies. The authors are themselves critical of these results and recommend the establishment of a uniform methodology for HFA testing with regard to age groups, comorbidities, hearing loss, gender and both occupational and leisure noise exposure. In terms of occupational exposure, information on the use of personal protective equipment should be included. The meta-analysis suggests that at 16 kHz, HFA is sensitive enough to identify early hearing loss in the CG. This is true for 4 kHz as well, but the outcome is not as significant. Further studies are therefore needed to validate the importance of using HFA to monitor early hearing loss in people exposed to workplace noise [31].

Our review compared the use of HFA in populations exposed to workplace noise in addition to comparing the EG and CG with regard to age categories, taking into account both CA and HFA. We found significant differences in the study methodologies used for HFA, mainly in terms of age group divisions. Some of the studies investigated did not consider particular age groups and evaluated hearing thresholds in HFA for one single group, regardless of the ages of the included individuals [18,19,20,22,28,29,30,31]. Previous studies have confirmed that HFA detects deterioration in the hearing threshold with an increasing age, increasing frequency and with noise exposure at younger ages compared with CA. A cross-sectional study of 6451 individuals with a mean age of 59 ± 4 years designed to be representative of the US population found a decrease in cognition with every 10 dB reduction in hearing, which continued below the clinical threshold such that subclinical levels of hearing impairment (below 25 dB) were significantly related to lower cognition [4,31]. Therefore, CA studies aimed at finding people with hearing loss should be extended to HFA studies with the same design.

### 3.3. Limitations

The limitations of this review article are as follows:Only a few published studies have used HFA to determine hearing thresholds in an unexposed population, and even fewer studies have compared cohorts exposed and unexposed to workplace noise;The different age groups considered in the existing studies have varying age ranges. Some studies have reported hearing thresholds independently of age, even though the hearing threshold worsens with age at each frequency;The standards for hearing loss classification and their corresponding audiometry values have not yet been defined for frequencies other than 0.125–8 kHz.

Due to these restrictions, it was not possible to perform a meta-analysis of the data. We could not compare the results of the individual studies with normative data. HFA needs to be studied further, especially with regard to dependence on age and other variables.

## 4. Conclusions

Until recently, it was not straightforward to examine hearing thresholds with HFA. HFA has made it into clinical practice mainly due to recent developments in the technology and the construction of special headphones and specially adapted audiometers. Although it can be used for the early detection of hearing loss, normative data (i.e., boundary values that indicate an as-yet unimpaired hearing threshold) have not yet been established. The effect of NIHL first became apparent through the use of HFA rather than CA. NIHL is a risk factor that is easily preventable in working environments, and moreover, it is highly underestimated as a risk factor in non-working environments in ordinary human life. Among people unexposed to workplace noise, significantly higher hearing thresholds are found with increasing age. The ability to detect sound is up to 18 kHz in those aged under 40 years, while it decreases to 14 kHz in the 40–49-year age group, and those above 50 years are able to detect sound only up to 11.2 kHz. Significantly higher hearing thresholds in EG compared with CG have been found in the 9–18 kHz range, especially at 16 kHz. Previous studies have also found higher hearing thresholds at 10–16 kHz in respondents aged under 31 years who have used PLDs for longer than 5 years. It is obvious that PLD users in younger age categories require more attention in terms of hearing loss detection, since these age categories are more familiar with modern technologies, including different types of PLDs.

The hearing threshold increases with increasing age, frequency and noise exposure, and this is detectable earlier when HFA rather than CA is used.

A number of studies on HFA have been conducted, but each had a different design, and so they cannot be reliably compared to other studies. A harmonized methodology needs to be established that takes into account variables such as age (e.g., WHO age categories), sex, comorbidities and noise exposure, as well as other socioeconomic and psychosocial factors and the usage of PLDs. This is necessary given the broader objective of earlier detection of hearing loss and compensation for this impairment. According to this context and the development of modern technologies, HFA should be added into the standard examination protocol for the early detection of hearing loss.

## Figures and Tables

**Figure 1 ijerph-18-04702-f001:**
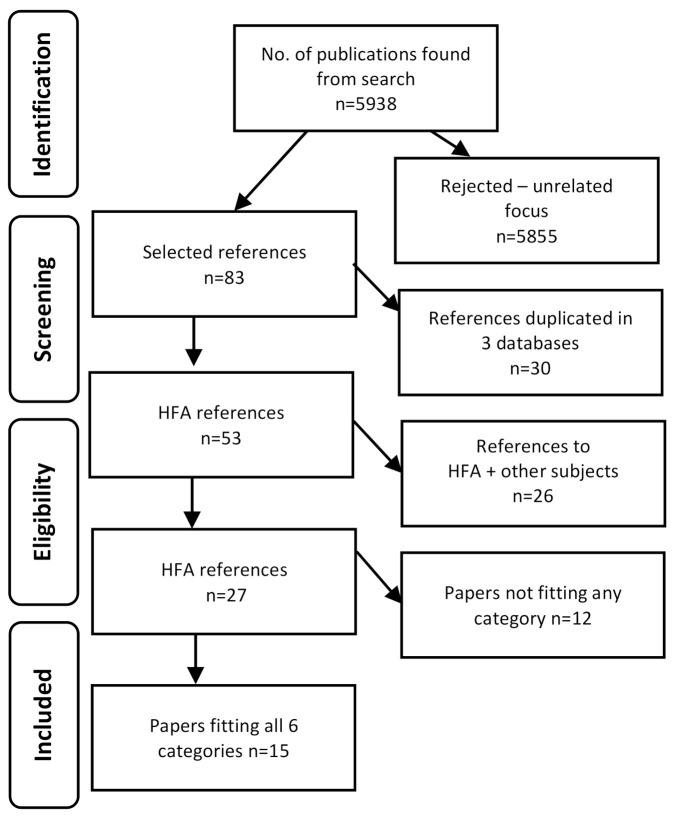
Flow chart of items used for the narrative review.

## Data Availability

The study materials and the details of all analyses are available from the corresponding authors upon reasonable request.

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
