# Peer review of "High-Frequency Audiometry for Early Detection of Hearing Loss: A Narrative Review"

_ijerph, 2021, doi:10.3390/ijerph18094702_

Round 1
Reviewer 1 Report
Thank you for submitting this manuscript. There are a few grammatical errors but these were minor.
I don't see how this contribution is really any different than that of Antoniolo et al (2016). This is a simple statement of the results of a few studies with a limited sample size. I would have liked to see a statistics section showing the D' values and other measures to show why it can't be a meta analysis. As such, it provides no additional information to the literature. Why could I just not read Antonioli et al and be happy with that?
You have also not discussed all of the important acoustics issues which contribute to the higher variability in the measured thresholds in the higher frequency region. For example, you have not discussed the standing wave pattern in the occluded ear canal, the acoustics of ear canal geometry, and other calibration factors that would vary from person to person.
Line by line:
22: specify PLD since its the first usage
49 and 50: "compensated" is an unusual word... perhaps "addressed"
62: It has not been established that hearing loss contributes to cognitive decline. This is only a corelative relationship.
65: Change "clear" to "pure"
67: Delete the words "...the air via..."
69: This is your conclusion. You can't conclude something that you havent yet addressed.
74: Delete "European". This journal is read in a number of international jurisdictions
76: Change "tone" to "puretone" and going forward (eg. line 80)
78: Use passive... as in "A literature search was conducted..." rather than "We conducted..."
106-108: Delete this section, since you just said all this a few paragraphs earlier.
Where is your discussion of some of your limitations that you mention?
Reviewer 2 Report
The topic of this paper is of interest. However, I think that the paper needs major revision to get it into publishable form. The main problems are:
1) It is not clear how this review increases knowledge/understanding relative to the review of Antonioli et al. (2016).
2) The writing is not precise. The authors often talk about “differences” or “effects” or something being “highest” or “lowest” when it is not clear what is being compared to what.
3) The English need to be checked and corrected by a native speaker.
4) The authors do not attempt to evaluate the quality of the studies that are reviewed. Also, they do not present a critical analysis. They simply summarise what the authors of the articles stated.
5) The search strategy used to identify relevant papers was not adequate. Several relevant papers were not identified, for example:
- Fausti, S.A.; Erickson, D.A.; Frey, R.H.; Rappaport, B.Z. The effects of impulsive noise upon human hearing sensitivity (8 to 20 kHz). Scand. Audiol. 1981, 10, 21-29.
- Fausti, S.A.; Erickson, D.A.; Frey, R.H.; Rappaport, B.Z.; Schechter, M.A. The effects of noise upon human hearing sensitivity from 8000 to 20 000 Hz. J. Acoust. Soc. Am. 1981, 69, 1343-1347.
- Couth, S.; Prendergast, G.; Guest, H.; Munro, K.J.; Moore, D.R.; Plack, C.J.; Ginsborg, J.; Dawes, P. Investigating the effects of noise exposure on self-report, behavioral and electrophysiological indices of hearing damage in musicians with normal audiometric thresholds. Hear. Res. 2020, 395, 108021, doi:10.1016/j.heares.2020.108021.
6) The introduction should focus on the main topic of the paper: methods for the early detection of hearing loss and the importance of such early detection. There is no need to describe what an audiometer is or how it works.
7) In the end, it is not clear whether or not HFA is useful for early detection of noise-induced hearing loss. Either a clear statement about this should be made, or needs for further research should be identified.
I have marked up a hard copy of the paper with many corrections, comments and suggestions for re-wording. A scan of this has been uploaded with this review. Some specific points are:
All abbreviations (e.g. PLD) should be defined when first used, and then used consistently.
References are often given in author (date) format rather than the recommended numbered format.
It makes no sense to say that “HFA is a predictor for the early detection of noise-induced hearing loss”. A more reasonable statement would be “HFA can be used to detect early signs of NIHL” or “The effects of NIHL first become apparent in HFA rather than CA”, if eith of these is actually supported by the evidence.
It is too strong to say that hearing loss “can lead to lowered mental acuity”. A causal link has not yet been established.
The fact that hearing screening is cost-effective for children does not logically imply that it is cost-effective for adults.
Why was the search restricted to the years 2000-2020? This excludes some relevant papers, for example:
- Green, D.M.; Kidd, G.; Stevens, K.N. High-frequency audiometric assessment of a young adult population. J. Acoust. Soc. Am. 1987, 81, 485-494.
The text does not list any studies from the USA, but at least one of the studies in the Table is from the USA. There are also relevant studies from the UK.
It is claimed that “HFA is more sensitive to NIHL detection than CA”, but the evidence for this statement is not clear.

Round 2
Reviewer 1 Report
Thank you for addressing my queries. You still have not demonstrated why this study contributes anything to the literature that was not done in a similar study five years ago.
Reviewer 2 Report
The authors have responded appropriately to my previous comments. However the English needs editing by a native English speaker. For example, in this new sentence in the abstract: "The effect of noise-induced hearing loss (NIHL) first become apparent in HFA rather than CA.", this should either read "... effect ... bcomes..." or "... effects ... become...".
